**ə | Open Peer Review** | Biotechnology | Research Article

# Engineering of chimeric enzymes with expanded tolerance to ionic strength

Paweł Mitkowski,[1,2] Elżbieta Jagielska,[1,2] Izabela Sabała[1,2]

**ABSTRACT** Antimicrobial resistance poses a significant global threat, reaching dangerously high levels as reported by the World Health Organization. The emergence and rapid spread of new resistance mechanisms, coupled with the absence of effective treatments in recent decades, have led to thousands of deaths annually from infections caused by drug-resistant microorganisms. Consequently, there is an urgent need for the development of new compounds capable of combating antibiotic-resistant bacteria. A promising class of molecules exhibiting potent bactericidal effects is peptidoglycan hydrolases. Previously, we cloned and characterized the biochemical properties of the M23 catalytic domain of the EnpA ($EnpA_{CD}$) protein from *Enterococcus faecalis*. Unlike other enzymes within the M23 family, $EnpA_{CD}$ demonstrates broad specificity. However, its activity is constrained under low ionic strength conditions. In this study, we present the engineering of three chimeric enzymes comprising $EnpA_{CD}$ fused with three distinct SH3b cell wall-binding domains. These chimeras exhibit enhanced tolerance to environmental conditions and sustained activity in bovine and human serum. Furthermore, our findings demonstrate that the addition of SH3b domains influences the activity of the chimeric enzymes, thereby expanding their potential applications in combating antimicrobial resistance.

**IMPORTANCE** These studies demonstrate that the addition of the SH3b-binding domain to the $EnpA_{CD}$ results in generation of chimeras with a broader tolerance to ionic strength and pH values, enabling them to remain active over a wider range of conditions. Such approach offers a relatively straightforward method for obtaining antibacterial enzymes with tailored properties and emphasizes the potential for proteins' engineering with enhanced functionality, contributing to the ongoing efforts to address antimicrobial resistance effectively.

**KEYWORDS** EnpA, SH3b, M23, bacteriolytic enzymes, chimera, *S. aureus*, *E. faecalis*

Antimicrobial resistance (AMR) is a worldwide problem accelerated by the inappropriate use and overconsumption of antibiotics across human healthcare, agriculture, and industry. The World Health Organization has identified bacterial species, termed "priority pathogens," that pose the greatest threat to human health. These pathogens, *Enterococcus faecium*, *Staphylococcus aureus*, *Klebsiella pneumoniae*, *Acinetobacter baumannii*, *Pseudomonas aeruginosa,* and *Enterobacter* spp., collectively known as ESKAPE, were accountable for 929,000 deaths attributed to AMR and 3.57 million deaths associated with AMR globally in 2019 (1, 2). Over the past three decades, only two new classes of antibiotics have been approved for human treatment (3). Consequently, there is an urgent need to intensify research efforts aimed at discovering new compounds and innovative strategies to combat bacterial infections effectively.

Peptidoglycan hydrolases (PGHs) represent a highly promising group of molecules with potent bactericidal effects. These enzymes cleave bonds within bacterial cell walls

Address correspondence to Izabela Sabała, isabala@imdik.pan.pl.

The authors declare no conflict of interest.

See the funding table on p. 14.

leading to instant death of bacteria (4). Peptidoglycan (PG) serves as the primary polymer in bacterial cell walls, comprising sugar moieties and amino acids that form a mesh-like structure encasing the cells. The sugar chain contains alternating residues of β-(1,4)-linked N-acetylglucosamine and N-acetylmuramic acid. Murein peptides, attached to N-acetylmuramic acid, establish connections between the sugar chains. The composition of PG, particularly its peptide components, is characteristic of specific bacterial genera or species. However, it can undergo alterations based on environmental conditions or the bacterial growth phase (5). For instance, *S. aureus* or *Enterococcus faecalis* contain a murein peptide with the following composition: L-Ala-D-iGlu-L-Lys-D-Ala-D-Ala and the cross-linking peptide: Gly-Gly-Gly-Gly-Gly and L-Ala-L-Ala, respectively (6, 7).

PGHs typically exhibit a modular architecture consisting of well-defined catalytic and binding domains (8). EnpA, originating from bacteriophage and found in the genome of *E. faecalis*, is a large multidomain protein that comprises a phage-related tail protein domain, murein D,D-endopeptidase MepM, murein hydrolase activator NlpD containing a LysM domain, peptidase M23 and lytic transglycosylase-like domain. M23 domains, categorized as zinc metallo-endopeptidases, play a pivotal role in cleaving bonds within PG (9). In our previous studies, we demonstrated that the isolated M23 domain of EnpA (EnpA$_{CD}$) exhibits potent lytic activity against a broad spectrum of bacterial species, including its host *E. faecalis* as well as some staphylococci and streptococci, but its activity is limited to low ionic strength (10).

In multidomain PGHs, catalytic domains are typically accompanied by cell wall-binding domains, which facilitate the enzyme attachment to the cell surface. They belong to various families, with SH3b, LysM, ChBD, and Clp7 being among the most common ones (11). Studies have demonstrated that these domains recognize distinct elements within bacterial cell walls. For instance, the LysM domain binds to glycan chains of PG (12), while SH3b domain, found in enzymes such as lysostaphin and Ale-1, recognizes pentaglycine cross-bridges present in staphylococcal PG (13, 14).

In our study, we have successfully engineered chimeric enzymes comprising EnpA$_{CD}$ and SH3b domains sourced from three distinct staphylococcal PGHs. These chimeras effectively retained the specificity of EnpA$_{CD}$ while demonstrating enhanced activity across an expanded range of ionic strength and pH conditions. With these improved characteristics, the chimeric enzymes hold significant potential for diverse applications as a novel class of nonantibiotic antibacterial agents. Furthermore, our analysis of the three chimeric enzymes, each featuring the same catalytic domain alongside different SH3b domains, has provided valuable insights into additional mechanisms governing interactions between SH3b domains and bacterial cell walls. This investigation sheds light on the diverse functionalities of SH3b domains beyond direct PG binding, further enriching our understanding of the molecular mechanisms underlying bacterial cell lysis and enzyme activity.

## RESULTS

### Chimera design and purification

In our previous research, we demonstrated that the fusion of the cell wall-binding domain (SH3b) with the LytM catalytic domain enables chimeric enzyme to maintain its activity even under higher ionic strength (15). To expand the activity of yet another M23 catalytic domain of EnpA$_{CD}$ to wide range of conditions, we applied the same approach. Three chimeric enzymes were engineered by fusing various SH3b domains and linker which occur naturally with this domains, to EnpA$_{CD}$ (GenBank ID: AE016830.1, protein ID: AAO81264.1, amino acids 1375–1501). The selected binding domains were sourced from lysostaphin, produced by *Staphylococcus simulans* (GenBank ID: U66883.1, protein ID: AAB53783.1, amino acids 387–493), and two other lysostaphin homologs produced by *S. simulans* (GenBank ID: LRQJ01000017.1, protein ID: KXA44996.1, amino acids 310–424) and *Staphylococcus pettenkoferi* (GenBank ID: CP022096.2, protein ID: ASE36562.1, amino acids 341–446) (Fig. 1). These three SH3b domains share 44% amino acid sequence identity within their 106 aa-long fragments but differ in pI and net charge (see Table S1).

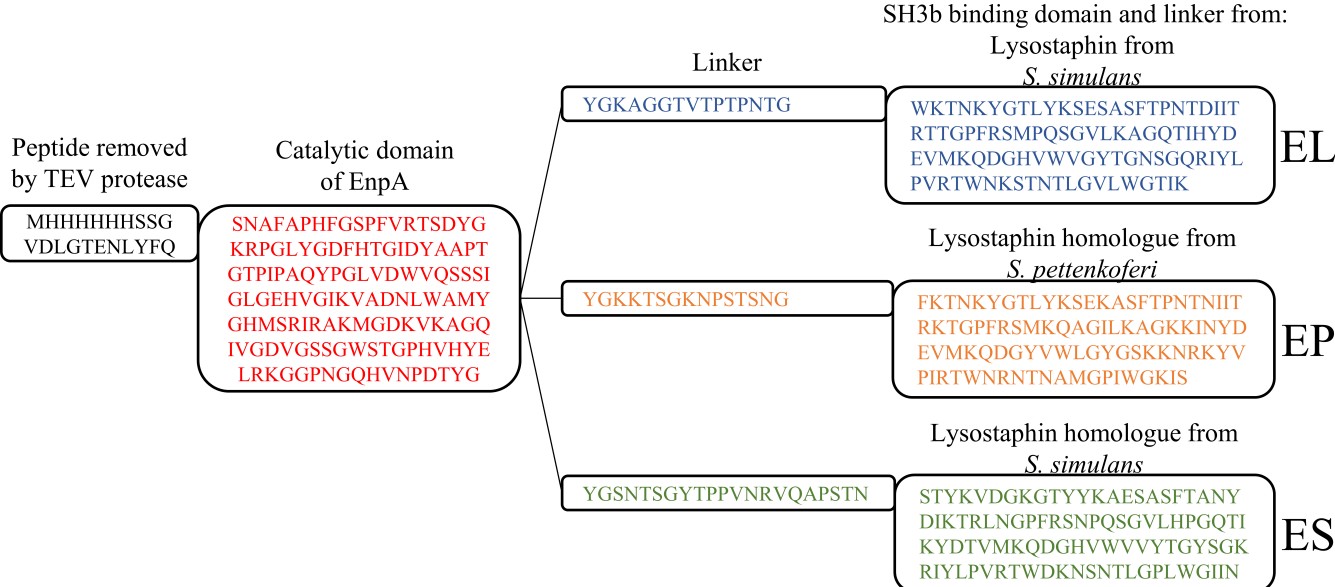

**FIG 1** Amino acid sequence of chimeric enzymes EL, EP, and ES. EnpA$_{CD}$ (GenBank ID: AE016830.1, protein ID: AAO81264.1, amino acids 1375–1501), SH3b from lysostaphin produced by *S. simulans* (GenBank ID: U66883.1, protein ID: AAB53783.1, amino acids 387–493), lysostaphin homolog produced by *S. simulans* (GenBank ID: LRQJ01000017.1, protein ID: KXA44996.1, amino acids 310–424) and lysostaphin homolog produced by *Staphylococcus pettenkoferi* (GenBank ID: CP022096.2, protein ID: ASE36562.1, amino acids 341–446). The linker and the SH3b domain are derived from naturally occurring proteins, but the linker fragment has not been modified.

The chimeric enzymes derived from EnpA$_{CD}$ with the attached binding domains from lysostaphin, and M23 peptidases from *S. pettenkoferi*, and *S. simulans* are abbreviated as EL, EP, and ES, respectively. EnpA$_{CD}$ and chimeras were produced with a HisTag, which was cut-off by Tobacco Etch Virus protease, and as a result, three amino acids have been added to the initial sequence senine, asparagine, alanine (SNA). The molecular weight of each purified chimeric enzyme was verified using mass spectrometry (Fig. S1).

## Activity under different ionic strength and pH values

To determine the optimal ionic strength for lytic activity, we tested single catalytic domain of EnpA$_{CD}$ and compared to the activity of the chimeras using a turbidity reduction assay against *S. aureus* National Collection of Type Cultures (NCTC) 8325–4 and *E. faecalis* American Type Culture Collection (ATCC) 29212 in 50 mM glycine buffer at pH 8.0, supplemented with various concentrations of NaCl ranging from 0 to 500 mM (0.3–39.9 mS/cm). On the other hand, buffers with conductivity of 0.5 mS/cm were used to determine the optimal pH for enzyme activity by appropriate dilution of the buffering solution with water (Fig. 2). The lytic activity of single domain EnpA$_{CD}$ was hindered by the presence of 6 mM NaCl, being ceased entirely at 25 mM NaCl for both tested bacterial strains. Fusion of the SH3b domains sustained the lytic activity of all three chimeras, EL, EP, and ES, across the entire range of tested ionic strength conditions (Fig. 2A); however, their efficacy declined beyond 25 mM NaCl. Notably, chimeric enzymes' sensitivity to ionic conditions was clearly depended on the bacterial strain tested. While all three chimeras responded similarly to increasing salt concentrations if *S. aureus* was applied as a substrate, distinct differences in activity were observed among enzymes, when *E. faecalis* cell were investigated. In the latter experiment, ES chimera maintained stable high efficiency across all tested conditions (0–500 mM NaCl), whereas the activity of the EL started to decrease at 25 mM NaCl and continued to drop gradually along with increasing concentration of NaCl. The EP chimera, however, displayed the lowest activity among tested enzymes and exhibited optimal lytic efficacy in the presence of 100 mM

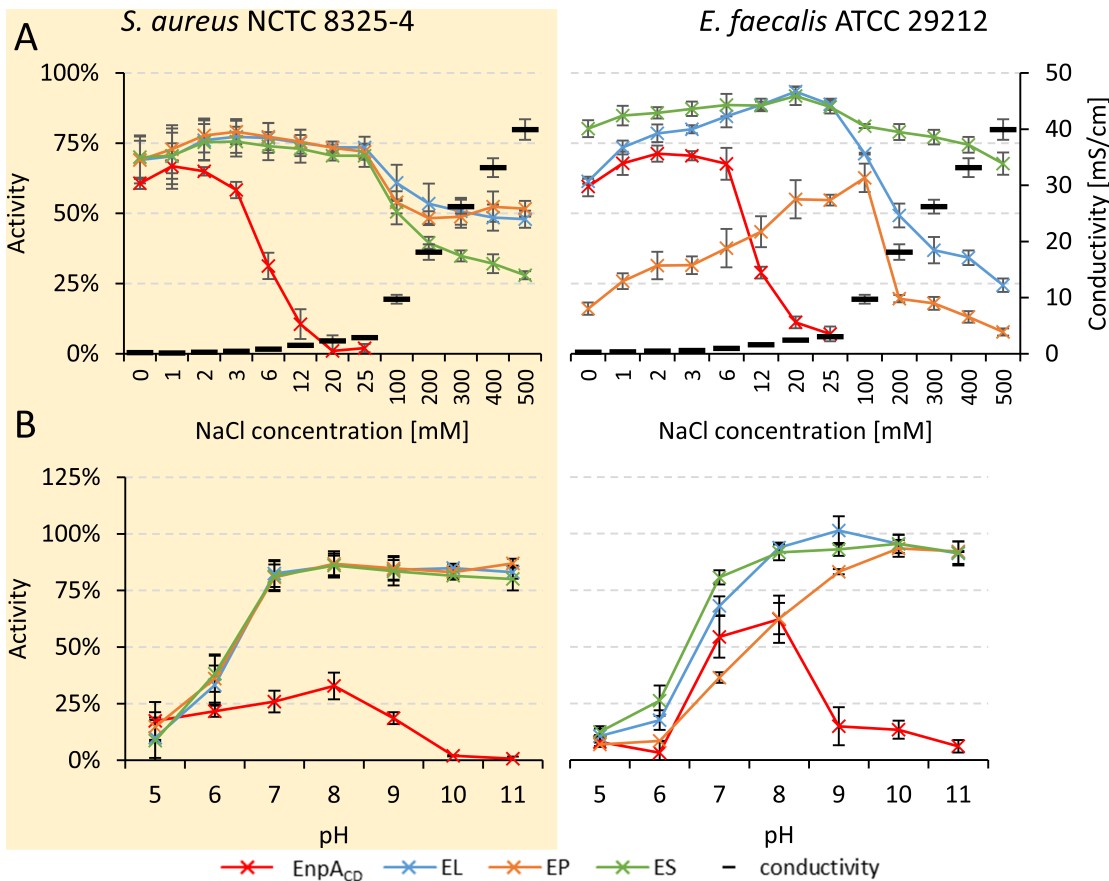

**FIG 2** The effect of conductivity (A) and pH values (B) on EnpA$_{CD}$ and chimeric enzymes activities. *S. aureus* NCTC 8325–4 and *E. faecalis* ATCC 29212 were incubated with 500 nM enzymes in various conductivity (0.1–39.9 mS/cm, pH 8.0) and pH conditions (constant conductivity 0.5 mS/cm) for 1 h at room temperature. The activity was calculated as a percentage of the reduction of the initial OD$_{620}$. The results were normalized to the negative control without added enzyme and presented as mean values and standard deviation derived from at least three technical and biological replications.

NaCl. In summary, the addition of SH3b domains expanded the range of ionic strength permissible for EnpA$_{CD}$ activity.

While the isolated catalytic domain showed activity within a narrow range of pH values, with an optimum pH between 7 and 8, all chimeric enzymes demonstrated lytic activity across a much broader pH range (7–11) (Fig. 2B). The remarkably high activity of all chimeric enzymes under basic conditions is particularly noteworthy, especially considering the complete inhibition of EnpA$_{CD}$ performance in solution of pH above 10 (in *S. aureus* tests) and 9 (in *E. faecalis* experiment). In conclusion, fusion of the cell wall-binding domain with catalytic domain not only broadened tolerance to pH conditions but also enhanced the activity of the chimeric enzymes while compared to the parental enzyme.

## Elimination of bacteria under low and high ionic strength

Given that turbidity reduction may not always directly reflect the actual efficiency of enzymes in eliminating bacterial cells, we conducted additional assessments by monitoring the survival of cells after enzyme treatment. The initial count of *S. aureus* NCTC 8325–4 and *E. faecalis* ATCC 29,212 cells was approximately 2.0 × 10$^7$ CFU/mL, and the surviving bacterial count was determined after 1 h of treatment with 500 nM enzymes at room temperature (Fig. 3). Under conditions of low ionic strength (50 mM glycine buffer without salt), both EnpA$_{CD}$ and all chimeras exhibited remarkably high activity against both strains. The bacterial cell count decreased from over 10$^7$ CFU/mL

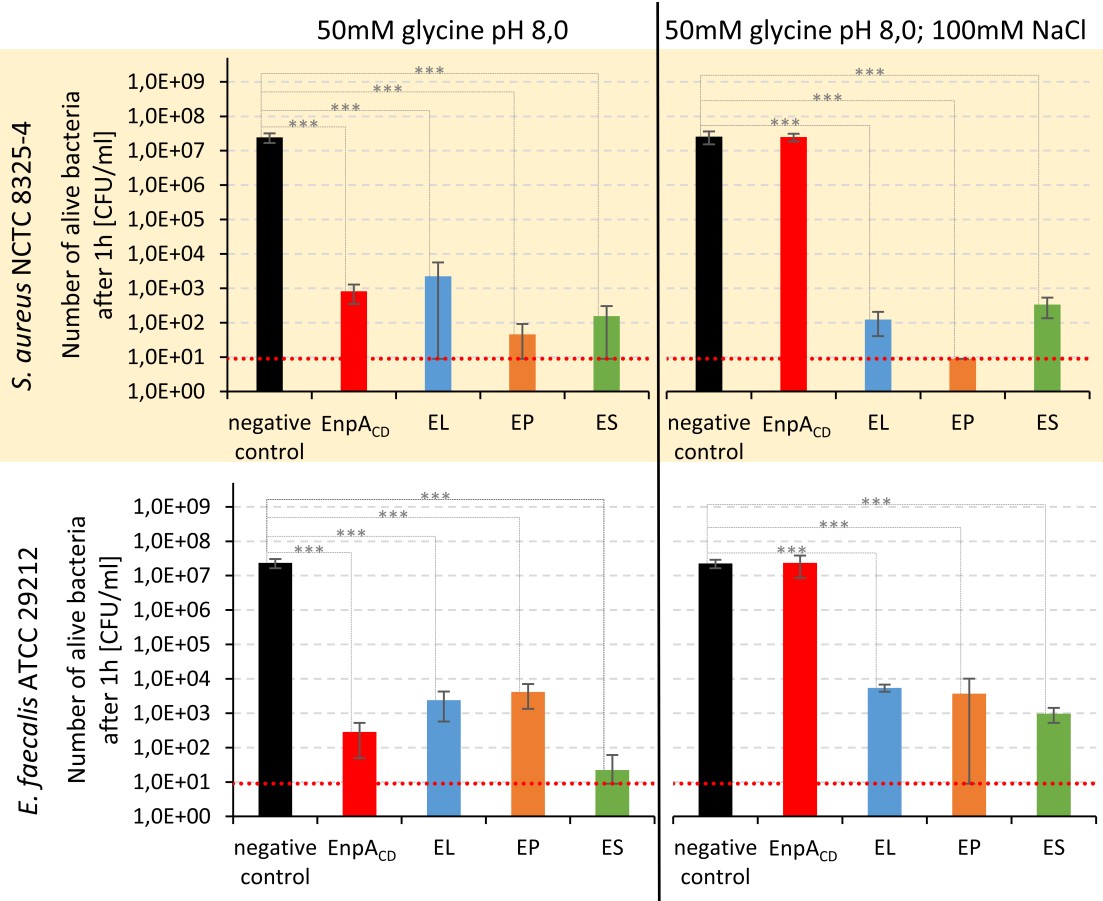

**FIG 3** Bacteriolytic potential of studied enzymes in buffer with low (no NaCl) and high (100 mM NaCl) ionic strength against *S. aureus* NCTC 8325–4 and *E. faecalis* ATCC 29212 incubated with 500 nM enzymes for 1 h at room temperature. The number of recovered cells is presented on logarithmic OY axis scale (mean values and standard deviation from three biological replicates). The dashed red line indicates the detection limit of the method. Statistical analysis - one-way analysis of variation (ANOVA) with post-hoc Scheffé multiple comparison test ($\alpha = 0.05$, $P$ value < 0.05 - *, <0.01 -**, <0.001 - ***).

by at least four logs, with the most significant reduction being six logs. As anticipated, EnpA$_{CD}$ activity diminished under conditions of higher ionic strength (50 mM glycine buffer with 100 mM NaCl), whereas the chimeric enzymes maintained their effectiveness. Statistical analysis revealed no significant differences in activity among the chimeras.

## Specificity against different bacterial strains

The specificity of EnpA$_{CD}$ was previously assessed across 32 bacterial strains with diverse PG structures and compositions (10). To investigate the role of the PG type in enzyme specificity, bacterial strains with diverse cross-bridge compositions were selected for the experiment. Furthermore, to evaluate how the addition of binding domains can expand enzymes tolerance to ionic strength, activity tests were conducted and under high-ionic strength conditions for chimeric enzymes (50 mM glycine buffer with 100 mM NaCl) (Fig. 4). In many cases, the chimeric enzymes exhibited higher activity levels against particular bacterial species compared to EnpA$_{CD}$ alone. For instance, the EP chimera displayed activity against *S. pettenkoferi* VCU012, whereas the other enzymes remained inactive. However, with *Streptococcus equi* subsp. *zooepidemicus*, only the EL and ES enzymes demonstrated high activity. In a few cases, the activity of the chimeric enzymes was significantly lower than that observed for parental catalytic domain EnpA$_{CD}$, for example, in tests of ES with *Staphylococcus saprophyticus* and EP with *Streptococcus agalactiae* just to mention the most spectacular (please see Fig. 4).

| White background - not statistically significant difference of activity | Green background - activity statistically significant higher than EnpA$_{CD}$ | * p<0.05 | ** p<0.01 | *** p<0.001 |
|---|---|---|---|---|
| | Red background - activity statistically significant lower than EnpA$_{CD}$ | * p<0.05 | ** p<0.01 | *** p<0.001 |

| Bacteria | Resources | Cross-bridge composition | EnpA | EL | EP | ES |
|---|---|---|---|---|---|---|
| *Micrococcus luteus* | ATCC 10240 | D-Ala-L-Lys-Glu(Gly)-L-Ala | 50±24% | 26±10% | 58±10% | 35±8% |
| *Lactococcus lactis* | CCM1877 | D-Asp | 7±4% | 5±4% | 10±4% | 15±4% |
| *Enterococcus faecium* | CCM 7167 | | 8±11% | 10±8% | 13±8% | 12±8% |
| *Bacillus Subtilis* | IBB 886 | direct | 10±1% | 17±5% | 23±5% | 16±6% |
| *Streptococcus oralis* | CCM 7412 | | 21±11% | 13±5% | 15±10% | 36±20% |
| *Aerococcus viridans* | CCM 1914 | | 78±5% | 66±15% | 51±3% | 62±7% |
| *Staphylococcus pettenkoferi* | VCU 012 | Gly$_{4-5}$-L-Ser$_{1-2}$ | 6±3% | 8±10% | 39±14% | 5±2% |
| *Staphylococcus pasteuri* | PCM 2445 | | 10±3% | 4±11% | 23±23% | 10±0% |
| *Staphylococcus pettenkoferi* | DSMZ 19554 | | 21±13% | 25±23% | 42±15% | 25±22% |
| *Staphylococcus capitis* | PCM 2121 | | 33±6% | 33±8% | 66±4% | 19±4% |
| *Staphylococcus warneri* | PCM 2107 | | 45±3% | 75±2% | 55±4% | 66±5% |
| *Staphylococcus haemolyticus* | PCM 2113 | | 46±13% | 49±7% | 88±5% | 51±8% |
| *Staphylococcus epidermidis* | ATCC 12228 | | 59±11% | 69±4% | 77±8% | 62±10% |
| *Staphylococcus simulans* | CCM 3583 | | 68±12% | 52±19% | 60±25% | 54±23% |
| *Staphylococcus intermedius* | DSM 20373 | | 71±11% | 76±5% | 74±7% | 66±13% |
| *Staphylococcus saprophyticus* | ATCC 15305 | | 72±10% | 43±8% | 83±9% | 39±6% |
| *Staphylococcus simulans* | DSMZ 20037 | | 82±7% | 71±12% | 84±2% | 76±9% |
| *Staphylococcus cohnii* | DSM 20260 | Gly$_{5-6}$ | 16±5% | 12±9% | 49±10% | 11±10% |
| *Staphylococcus xylosus* | PCM 2114 | | 21±3% | 27±2% | 67±2% | 21±0% |
| *Staphylococcus aureus* | NCTC 8325-4 | | 36±17% | 38±6% | 54±24% | 30±5% |
| *Staphylococcus agnetis* | DSMZ 23656 | | 37±15% | 69±6% | 56±9% | 60±7% |
| *Staphylococcus lugdunensis* | PCM 2430 | | 73±5% | 70±3% | 74±4% | 67±4% |
| *Salinicoccus roseus* | CCM 168 | | 81±16% | 86±6% | 73±4% | 85±5% |
| *Streptococcus equi* subsp. *zooepidemicus* | CCM 7316 | L-Ala$_{2-3}$ | 8±6% | 56±6% | 13±3% | 60±5% |
| *Streptococcus pyogenes* | CCM 7418 | | 33±5% | 57±8% | 21±5% | 59±8% |
| *Streptococcus ratti* | CCM 7443 | | 44±4% | 31±2% | 24±5% | 54±3% |
| *Enterococcus faecalis* | ATCC 29212 | | 60±1% | 71±4% | 66±4% | 81±6% |
| *Enterococcus faecalis* | CCM 1875 | | 64±3% | 52±6% | 40±8% | 61±4% |
| *Enterococcus faecalis* | CCM 4224 | | 69±2% | 56±7% | 42±5% | 65±7% |
| *Streptococcus agalactiae* | IBB 130 | | 67±5% | 66±5% | 32±10% | 70±2% |

**FIG 4** Comparison of the bacteriolytic potential of EnpA$_{CD}$ (tested at low ionic strength) and chimeras (tested at high ionic strength) against different bacterial strains determined by a turbidity reduction assay. The bacteriolytic potential was calculated as the reduction at initial OD$_{620}$ expressed as a percentage of the control (no enzyme) after 1 h of incubation at room temperature. The table shows the mean value and standard deviation from at least three technical and biological replications. (*P* value < 0.05 - *, <0.01 -**, <0.001 - ***, one-way ANOVA with post-hoc Scheffé multiple comparison test).

Bacteria with D-Asp in the PG cross-bridge were least susceptible to all enzymes among tested strains. For bacteria with direct cross-bridges, no significant differences were observed, apart from the EP chimera that displayed reduced activity against *Aerococcus viridans*. The effects were diverse for strains with glycine-serine cross-bridges. The EP chimera exhibited notably higher activity against *Staphylococcus capitis*, *Staphylococcus epidermidis*, *Staphylococcus haemolyticus*, and *S. pettenkoferi* VCU012.

In contrast, the EL and ES chimeras demonstrated increased activity against *Staphylococcus warneri* but lower activity against *Staphylococcus saprophyticus*. In the case of bacteria with polyglycine cross-bridges, the activity of all chimeric enzymes matched or surpassed that of EnpA$_{CD}$. Notably, the chimeras exhibited significantly higher activity, especially against strains such as *Staphylococcus cohnii*, *Staphylococcus xylosus*, or *Staphylococcus agnetis*.

The fusion of the SH3b domain yielded the most diverse effects in bacteria with L-Ala in the cross-bridge. For instance, the EL and ES chimeras exhibited high activity against *Streptococcus equi* subsp. *zooepidemicus* and *Streptococcus pyogenes*. In contrast, the EP chimera demonstrated lower activity against most species within this group that were tested.

In accordance with the results obtained, several general conclusions can be drawn: (i) In the majority of cases, the activity level of the generated chimeras in a high ionic strength buffer was either higher or maintained at the same level as the separate catalytic domain in a low ionic strength buffer; (ii) The activity level of the chimera against bacteria with the same murein type varied widely, indicating that the specificity of the chimeric enzymes was influenced by other features of the bacterial cell wall; (iii) In most instances, the EL and ES chimeras displayed similar effects, which were typically opposite to the effects observed for the EP chimera.

## Inhibition of enzyme activity

Bacteriolytic enzymes intended for practical use must exhibit high tolerance to diverse environmental conditions. Moreover, understanding methods for enzyme deactivation is crucial. While the addition of ion chelators is commonly employed to inhibit enzymes with metal ions as cofactors, thermal deactivation represents a more universal approach.

In tests conducted with *S. aureus* and *E. faecalis*, EnpA$_{CD}$ lost its activity in the presence of 5 mM EDTA, while the chimeras maintained their activity even at higher concentrations of EDTA. Moreover, the chimeric enzymes exhibited increased sensitivity to EDTA when tested against *S. aureus*. Surprisingly, when treating *E. faecalis*, the EP chimera displayed optimal activity in the presence of 10 mM EDTA (Fig. 5A). Notably, to completely block the activity of chimeras, a very high concentration of EDTA, up to 100 mM, is required.

These observations highlight the differential response of EnpA$_{CD}$ and chimeric enzymes to EDTA, indicating variations in their dependence on metal ions for catalytic activity. Furthermore, the chimeras' ability to maintain activity at higher EDTA concentrations underscores their robustness and potential utility in practical applications.

The thermostability of the enzymes was assessed by subjecting them to heating for 10 min at temperatures ranging up to 100°C. Results indicated that both EnpA$_{CD}$ and the EP chimera retained their activity even after treatment at the highest temperature. However, the EL chimera exhibited partial inactivation, while the ES chimera was completely inactivated at temperatures exceeding 60°C (Fig. 5B). These findings suggest differential thermal stability among the tested enzymes, with EnpA$_{CD}$ and the EP chimera demonstrating greater resistance to high temperatures compared to the EL and ES chimeras. Understanding the thermostability profiles of these enzymes is crucial for their practical application in various environments and conditions.

## Activity in serum

Despite the efficient elimination of bacteria by chimeric enzymes in environments with higher ionic strength, their activity may not be assured under physiological conditions such as serum. The presence of various ions, lipids, and proteins in serum could potentially hinder enzymatic activity. Therefore, the bacteriolytic properties of the chimeras were evaluated against *S. aureus* and *E. faecalis* in the presence of human and fetal bovine serum (FBS) (Fig. 6).

All chimeric enzymes maintained their activity in both human and bovine serum against *S. aureus*. The EP chimera completely eradicated *S. aureus* after 1 h, while for EL

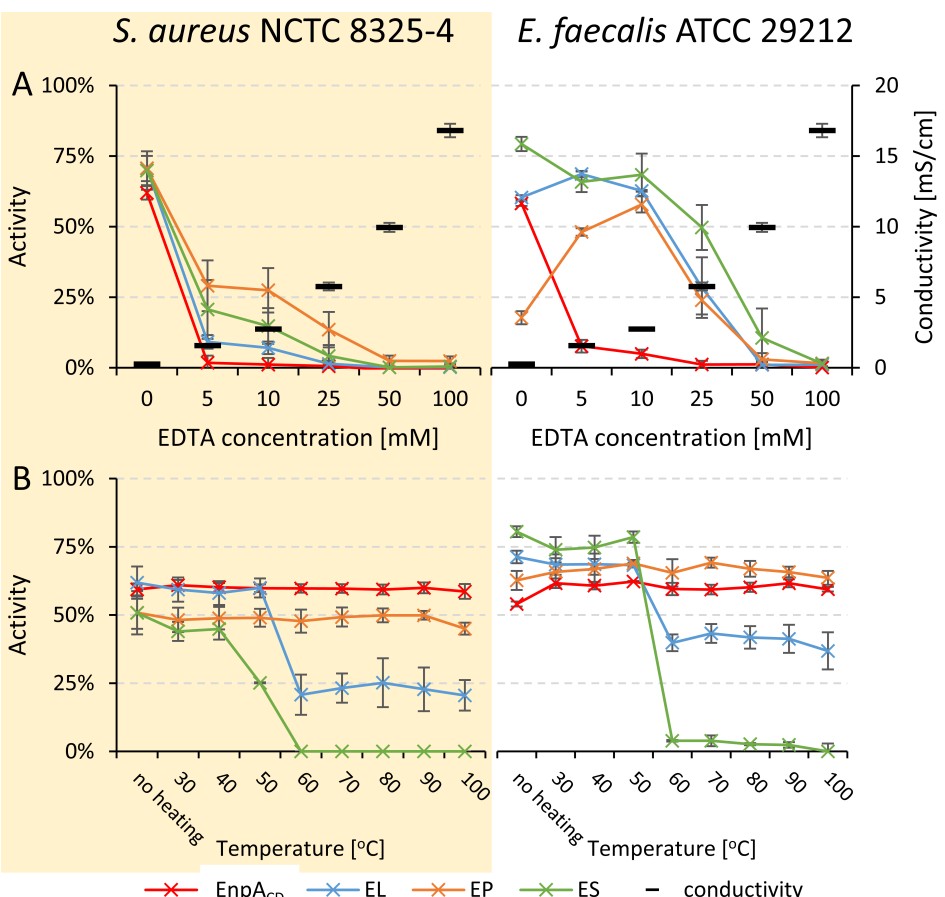

**FIG 5** Effect of EDTA (A) and heating (B) on the activity of chimeras. *S. aureus* NCTC 8325–4 and *E. faecalis* ATCC 29212 were incubated with 500 nM enzymes in 50 mM glycine buffer, pH 8.0 (additionally supplemented with 100 mM NaCl for chimeric enzymes) with increasing concentration of EDTA (0–100 mM, (A) or with preheated enzymes (30–100℃, (B). The activity is presented as the percentage of the reduction in initial $OD_{620}$ normalized to negative control (no added enzyme). Results are shown as mean value with standard deviation from three technical and biological replications.

and ES, optimal effects were observed after 3 h. However, in the case of ES, regrowth of bacterial cultures was observed after 24 h.

The activity of all chimeras against *E. faecalis* was generally lower compared to *S. aureus*. The ES chimera yielded the best results in this case, although recovery of bacterial cell growth was observed after 24 h in both sera. The EL chimera performed better in FBS than in human serum, where partial regrowth of bacterial cells was observed after 3 h. In contrast, the EP chimera did not exhibit any activity in FBS, while its antibacterial efficiency in human serum, albeit limited, was sustained throughout the experiment.

In summary, in serum environments, the chimeras demonstrate higher activity against *S. aureus*, than *E. faecalis*, with the EL and EP successfully eliminating all bacterial cells, as evidenced by the absence of regrowth after 24 h. *E. faecalis* bacteria, although reduced initially by the enzymes, are not eradicated completely under the tested conditions.

## DISCUSSION

The modular architecture of PGHs offers an opportunity to create chimeric enzymes by recombining individual domains from different proteins (16, 17). This approach can serve various purposes, such as increasing bacteriolytic activity, broadening the spectrum of activity against a greater number of bacterial species, or extending the half-life in the human body (18, 19). Chimeric enzymes can be generated through precise design based

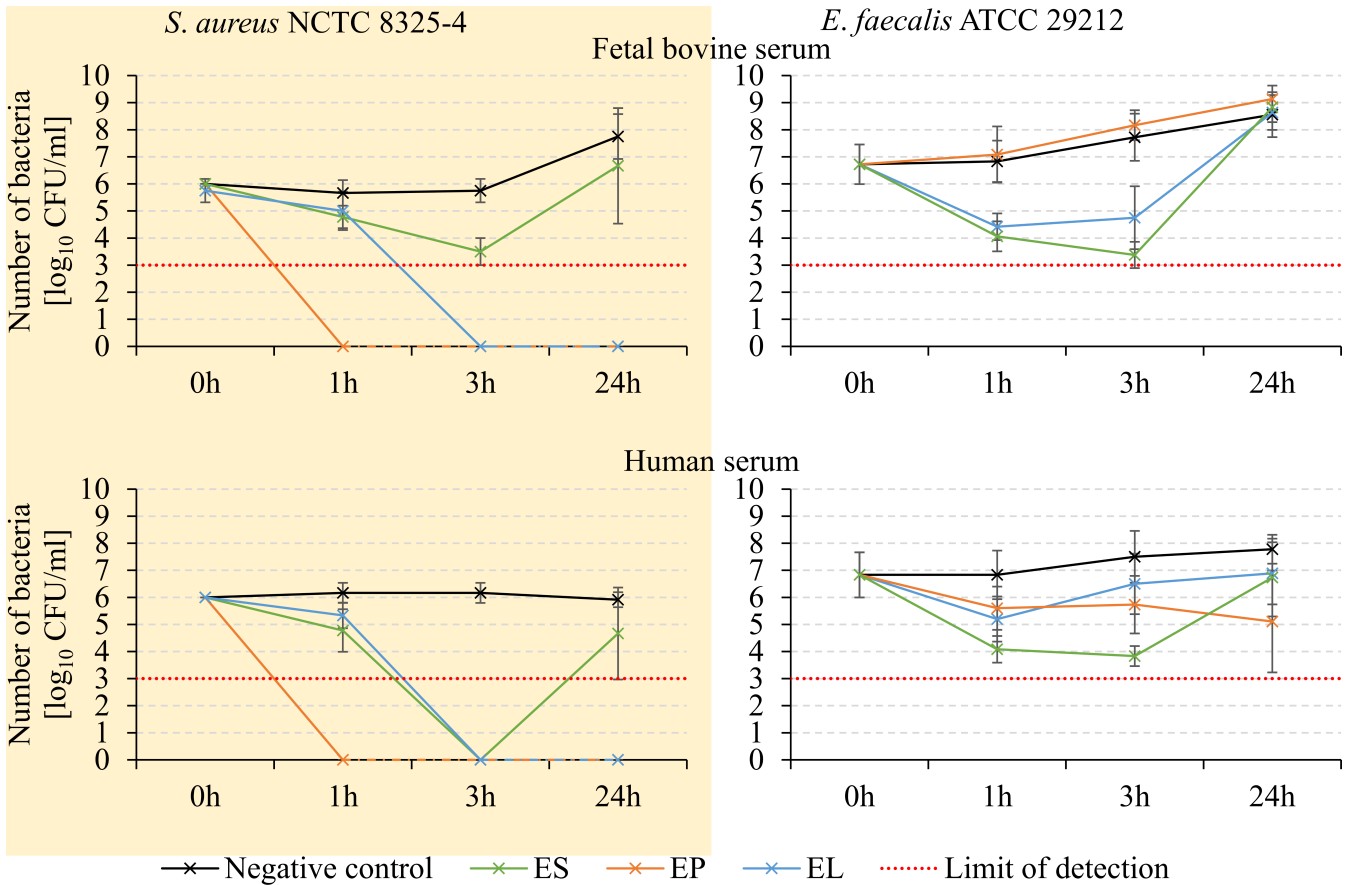

**FIG 6** Activity of chimeric enzymes in fetal bovine and human serum. *S. aureus* NCTC 8325–4 and *E. faecalis* ATCC 29212 were incubated in serum with 500 nM enzymes at room temperature for 1, 3, and 24 h. Results are presented as number or recovered bacterial cells (log$_{10}$ CFU/mL) and expressed as mean value with standard deviation from at least three technical and biological replications. The dashed red line indicates the detection limit of the method.

on structural and biochemical data or through random shuffling of numerous domains. For instance, the VersaTile high-throughput platform enabled the testing of almost 10,000 chimeras against *A. baumannii*. Remarkably, only five variants were found to be active against four tested strains in 90% human serum (20). This underscores the potential of chimeric enzymes in addressing specific bacterial targets and highlights the importance of innovative approaches in enzyme design and development.

Similar to other isolated catalytic domains of M23 peptidases (15), EnpA$_{CD}$ displayed high activity only under low ionic strength conditions. This phenomenon was also observed in the single catalytic domain of LysK (21), lysostaphin, and mature LytM (15). The primary objective of our protein engineering was to enhance the tolerance of EnpA$_{CD}$ to higher ionic strength by creating chimeric enzymes. These chimeras were designed based on detailed structural and biochemical data, building upon successful strategies employed in previous studies, where fusion of the LytM catalytic domain with the SH3b domain from lysostaphin resulted in chimeric enzymes with extended tolerance for ionic strength and pH values, while maintaining specificity (10, 15, 22). Similarly, the tolerance toward ionic strength and pH values was improved in the EnpA chimeras.

In general, the addition of the SH3b domain did not alter the specificity of the chimeric enzymes compared to EnpA$_{CD}$ indicating that the specificity of the chimera is determined by specificity of the catalytic domain; however, interesting observations were done regarding the role of the binding domains. In several cases, the performance of the chimeric enzymes surpassed that of EnpA$_{CD}$, while in very few cases, it was

inferior. Notably, the effects of the EL and ES chimeras were very similar but opposite to those of the EP chimera (Fig. 4).

While our understanding of the domains' specificity, structure, or binding mechanisms remains limited, and as such, cannot fully elucidate their effects on chimeric enzymes performance, we have made an effort to analyze the relationship between the features of the domains and the performance of the chimeric enzymes. To gain some insights, three-dimensional models of our chimeras were generated using the AlphaFold online tool (Fig. S2) (23). Despite careful analysis and alignment of the generated models with the crystal structure of lysostaphin, drawing relevant conclusions proved challenging. Only subtle differences in loop arrangements were observed, particularly in the most mobile regions. While differences in the mutual arrangement of both domains were noted, they could rise from unstructured character of linkers which are very flexible as demonstrated experimentally for lysostaphin (24). In conclusion, no apparent correlation between the predicted structures of SH3b-binding domains and their demonstrated properties as part of chimeric enzymes could be discerned.

The SH3b domain has been proposed to function as an anchor for the catalytic domain, facilitating enhanced affinity to bacterial cell walls (25). Our previous structural and biochemical studies have shown that the selectivity of the lysostaphin SH3b domain is intricately linked to the architecture of the binding groove, specifically designed to accommodate pentaglycine within staphylococcal PG (26). However, it does not fully account for the enhanced performance of the EL chimera, particularly against bacteria lacking pentaglycine in their PG cross-bridges. One plausible explanation is that this SH3b domains may also interact with the stem peptide of bacterial PG, a conserved feature among various bacterial species. This interaction could contribute to the broader efficacy observed in the EL chimera across different bacterial strains. Recently, we have reported that a domain from *S. pettenkoferi* SpM23B protein demonstrates efficient binding to a wide array of bacterial cells, including Gram-negative species suggesting that SH3b domains might interact with cell wall elements other than PG (27, 28).

Recent findings have highlighted the significant role of electrostatic interactions in mediating enzyme/cell wall interactions, shedding light on the mechanisms underlying bacteriolytic activity (27, 28). Studies have indicated that the positive surface charge of the catalytic domain of PGHs enables them to maintain their bacteriolytic activity even after the removal of the binding domain (29). Moreover, increasing the charge of the binding domain has been associated with enhanced bacteriolytic activity (30). In the case of the generated chimeric enzymes, their features were analyzed using Protein PI online software (31), revealing similar pI values ranging from 9.37 to 9.79. Interestingly, their calculated surface net charge was found to be similar for ES and EL (6.78 and 6.72, respectively), but significantly different for EP (17.67) (Supplemental data Table S1). This variation in surface net charge may contribute to the observed differences in the activities of ES and EL, as well as their opposite effects compared to EP. However, further research is necessary to confirm these conclusions and elucidate the exact mechanisms underlying these observations. One of the challenges in this analysis lies in the heterogeneity of bacterial cell walls and their variability across different species. Understanding the nuances of enzyme-cell wall interactions requires comprehensive investigation and consideration of various factors influencing bacterial susceptibility to enzymatic degradation.

The pH value also affects the surface charge of both proteins and the bacterial cell wall (32, 33). EnpA chimeras exhibit high effectiveness at neutral and alkaline pH but lose their activity at acidic pH due to the absence of direct interactions between protonated histidine and zinc ions at acidic pH (34). A similar phenomenon also occurs with the autolysin AtlE, which becomes inactive in acidic environments below 6.5 (35). In case of AtlA enzyme, this loss of activity could be attributed to protons in the respiratory chain being captured by teichoic acids, leading to local acidification of the environment and inhibition of enzyme activity (36).

The observed differences between chimeras in their response to heat treatment are intriguing. Chimera ES exhibited remarkable resistance to heat and maintained activity even after 10 min of heating at 100°C. In contrast, the EP chimera completely lost activity at 60°C. Interestingly, in a low ionic strength buffer, all chimeras retained their high activity even after heating at 100°C. This suggests that the loss of activity was primarily due to the detrimental effect of the treatment on the SH3b domain rather than the catalytic domain. Additionally, it has been verified that during heating, the chimeras do not break down into single domains, indicating the structural integrity of the chimeric enzymes even under harsh conditions (Fig. S3).

Another interesting observation concerns the increase in EP chimera activity with the increase in EDTA concentration, up to 10 mM and above this concentration, the activity decreases in similar manner like other chimeras. The explanation may be the fact that EP chimer is most active at a particular ionic strength. Also in the case of the EnpA$_{CD}$, its rapid loss of activity is related to the increase in the ionic strength of the solution rather than the action of EDTA (Fig. 2A; Fig. 5A results for *E. faecalis*).

By fusing various cell wall-binding domains to EnpA M23 catalytic domain, we have generated chimeric enzymes with potent bacteriolytic activity expanded to physiological conditions. The development of PGHs as therapeutic agents represents a promising alternative strategy for addressing AMR. PGHs have shown efficacy in reconstituting the balance of skin microflora in various skin disorders, and they are being integrated into dermocosmetics by companies like Micreos. Moreover, PGHs are undergoing clinical trials for systemic treatment of conditions like bacteremia and *S. aureus* endocarditis. The engineered chimeras discussed in our research were specifically designed to function effectively under conditions of high ionic strength and have demonstrated activity in both human and bovine serum. As a result, these chimeras hold significant potential as novel agents against polymicrobial infections caused by different pathogenic bacteria, including *S. aureus* and *E. faecalis*. This versatility and efficacy make them promising candidates for addressing antimicrobial challenges in both topical and systemic applications.

## MATERIALS AND METHODS

### Engineering of EnpA$_{CD}$ and chimeric enzymes

The Polymerase Incomplete Primer Extension (PIPE) cloning method was used to prepare the EnpA$_{CD}$ and chimeras (37). For this purpose, vector (EnpA$_{CD}$ in pMCSG7) (38) and insert amplification (linker and SH3b domain) were prepared using specific primers (Table 1). The resulting construct contained the following components (from the N-terminus): HisTag, Tobacco Etch Virus (TEV) protease cleavage site, EnpA catalytic domain and for chimeras additional SH3b domain with its linker and stop codon. After TEV treatment, three amino acids were added to the N-terminus (serine, asparagine and alanine). The PCR program: 10 µL 5× Phusion Hydrofluoric acid(HF) buffer, 1 µL

**TABLE 1** Primers' sequences used for EnpA$_{CD}$ and chimeras cloning

|  |  | 5′- > 3′ sequence |
| --- | --- | --- |
| Vector | Forward | ATAAGTATCAGGATTTACGTGTTGGCCATTTGGTCCGCC |
|  | Reverse | TAAGGCGATACCATAAATTCGAGCTCCGTCGACAAGCTTGCG |
| EnpA$_{CD}$ | Forward | TACTTCCAATCCAATGCCTTTGCTGCACATTTTGGATCACCATTGTT |
|  | Reverse | TTATCCACTTCCAATGTTAGCCATAAGTATCAGGATTTACGTGTTGG |
| Insert SH3b from Lysostaphin | Forward | AATCCTGATACTTATGGATATGGAAAAGCAGGTGGTACAGTA |
|  | Reverse | TATGGTATCGCCTTACTTTATAGTTCCCCAAAGAACACCTAA |
| Insert from SpM23_B | Forward | AATCCTGATACTTATGGTTATGGTAAAAAAACTAGTGGTAAG |
|  | Reverse | TATGGTATCGCCTTAACTAATTTTCCCCCAAATTGGTCCCAT |
| Insert form LssR_M | Forward | AATCCTGATACTTATGGATATGGTAGTAATACTTCTGGTTAT |
|  | Reverse | TATGGTATCGCCTTAATTAATAATTCCCCATAACGGGCCTAA |

10 mM dNTP, 0.5 µM (final concentration) of forward and reverse primers, 20 ng of DNA matrix, and 0.5 µL Phusion High-Fidelity DNA polymerase (Thermo Fisher Scientific). PCR program: 98°C – 30 s, (98°C – 7 s, 55°C – 20 s, 72°C – 90 s) x35 cycles, 72°C – 300 s, 18°C – hold. The PCR products were treated with DpnI (EureX) for 1 h, mixed together in a 1:3 molar vector:insert ratio and heated to 98°C for 5 min to hybridize them. The obtained constructs were sequenced.

## Protein expression and purification

Sequence-confirmed plasmids were used to transform *Escherichia coli* BL21 (DE3). Protein expression was carried out in autoinduction medium LB (AIM-LB, Formedium UK) with ampicillin at 25°C overnight with shaking at 180 revolutions per minute(RPM). Bacteria with overexpressed protein were suspended in 50 mM HEPES pH 8.0, 1 M NaCl, 10% glycerol, and 20 mM imidazole (buffer A). The sample was sonicated on ice for 5 min (cycle: 15 s of work, 60 s of rest) and centrifuged (20000 relative centrifugal force (RCF), 30 min, 4°C), and the supernatant with soluble protein was applied to a HisTrap fast flow (FF) 1 mL column (Cytiva). After washing the column with buffer A, the recombinant protein was eluted with 20 mM HEPES pH 8.0, 0.4 M NaCl, 10% glycerol, and 500 mM imidazole and dialyzed overnight in the presence of TEV protease at room temperature in 20 mM HEPES pH 8.0, 0.4 M NaCl, and 10% glycerol (buffer B). The protein solution was reapplied to the HisTrap FF 1 mL column, and the enzyme without HisTag was collected in flow through. Size exclusion chromatography on a HiLoad 16/600 Superdex 75 pg (Cytiva) column using buffer B was carried out as the last step of protein purification. The pure protein was flash frozen in liquid nitrogen and stored at –80°C. Molecular weight of each purified protein was confirmed by mass spectrometry.

## Bacteria preparation for tests

The bacteria from the glycerol stock were streaked on Tryptone Soya Broth (TSB)-agar plates and grown overnight at 37°C. A single colony was grown overnight in TSB medium at 37°C with shaking at 80 RPM and used as 1% inoculum in fresh TSB medium. The bacteria were grown at 37°C with shaking at 80 RPM and collected after reaching a log phase, $OD_{620}$ of 0.6–0.8 (using TECAN Infinite F50 Plus) by centrifugation (3500 RCF, 10 min, 20°C).

## Enzyme activity—determination of number of survived cells (CFU/mL)

The test was carried out in 50 mM glycine-NaOH pH 8.0 (for $EnpA_{CD}$) and with additional 100 mM NaCl (for chimeras). The pelleted bacteria were suspended in reaction buffer (same as enzyme) to obtain ca. $5.0 \times 10^7$ CFU/mL. Then 100 µL of the bacterial suspension was mixed with 100 µL of the enzyme solution (final enzyme concentration 500 nM). After 1 h of incubation at RT, 10-fold serial dilutions were made for each sample. One hundred microliters of each dilution was spread on TSB-agar plates, and after an overnight incubation at 37°C, number of colony was counted. Accordingly, the colony forming unit per milliliter (CFU/mL) was calculated. The limit of detection using this method is $10^1$ CFU/mL.

## Enzyme activity in serum

Tested bacteria and enzymes were suspended in human or bovine serum. The pelleted bacteria were suspended in serum to obtain ca. $5.0 \times 10^6$ CFU/mL. Then 100 µL of the bacterial suspension was mixed with 100 µL of the enzyme solution (final enzyme concentration 500 nM). Enzyme activity was determined at time 0 and after 1, 3, and 24 h incubation at room temperature by serial 10-fold dilutions. In each time points, 1 µL drop from each serial dilution was placed on a TSB-agar plate and incubated overnight at 37°C. After this time, it was checked in how many drops from the dilution series, the bacteria had grown, and based on this, the number of bacteria in the orders of

magnitude that survived was determined. On the basis of that, the $\log_{10}$ CFU/mL was calculated. The limit of detection using this method is $3\log_{10}$ CFU/mL.

### Enzyme activity—turbidity reduction assay

The pelleted bacteria and enzymes were suspended in the same buffer (the compositions of the buffers in the individual tests are described below). In each well of the 96-well plate, 100 µL of the bacterial suspension ($OD_{620}$ during test 1.0) was mixed with 100 µL of the enzyme solution (enzyme concentration during test 500 nM). $OD_{620}$ of the reaction was monitored for 1 h using TECAN Infinite F50 Plus with 2-min intervals. The enzyme activity was determined as a percentage of the reduction of initial $OD_{620}$ and normalized to negative control (without added enzyme).

### Effect of heating on enzyme activity

In the assayenzymes were diluted in 50 mM glycine buffer pH 8.0 with 100 mM NaCl (chimeras) or without salt ($EnpA_{CD}$). Then the enzyme solutions were heated for 10 min in a thermoblock at: 30.0, 40.0, 50.0, 60.0, 70.0, 80.0, 90.0 and 100.0°C. The samples were cooled at room temperature and a turbidity reduction assay was performed as described above. Final enzyme concentration was 500 nM and initial $OD_{620}$ of bacteria was 1.0.

### Dependence of enzymatic activity on ionic strength

The turbidity reduction test was carried out with 500 nM enzymes in 50 mM glycine-NaOH buffer, pH 8.0, with the following final NaCl concentrations: 0, 1.0, 2.0, 3.0, 6.0, 12.0, 25.0, 100.0, 200.0, 300.0, 400.0, and 500.0 mM; these prepared buffer had the following conductivity values given in (mS/cm): 0.3 ± 0.1; 0.4 ± 0.1; 0.5 ± 0.1; 0.6 ± 0.1; 1.0 ± 0.1; 1.6 ± 0.1; 2.4 ± 0.1; 3.0 ± 0.2; 9.7 ± 0.8; 18.1 ± 1.4; 26.2 ± 1.2; 33.1 ± 1.7; 39.9 ± 1.8, respectively. Final enzyme concentration was 500 nM and initial $OD_{620}$ of bacteria was 1.0.

### Dependence of enzymatic activity on pH

The turbidity reduction test was carried out in different buffers: citric pH 5.0 and 6.0, Tris-HCl pH 7.0, 8.0, and 9.0, CAPS pH 10.0 and 11.0. Buffers' concentration was adjusted (by proper dilution with milliQ water) to have the same conductivity of 0.5 mS/cm. Final enzyme concentration was 500 nM and initial $OD_{620}$ of bacteria was 1.0.

### Dependence of enzymatic activity on different EDTA concentrations

The turbidity reduction test was carried out in 50 mM buffer glycine-NaOH, pH 8.0 with the following EDTA final concentrations: 0, 5.0, 10.0, 25.0, 50.0, 100.0 mM. Final enzymes' concentration was 500 nM and initial $OD_{620}$ of bacteria was 1.0.

### Statistical analysis

The following method was used for statistical analysis, one-way ANOVA with post-hoc Scheffé multiple comparison test. Significance level used to compute the confidence level was $\alpha = 0.05$.

### Mass spectrometry analysis

The molecular weight measurement was performed on a quadrupole time-of-flight (Q-TOF) Premier spectrometer from Waters company. The deconvolution of the obtained spectra (m/z) was performed using software MassLynx Mass Spectrometry Software from Waters company. The MaxEnt 1 tool was used to calculate the mass. MaxEnt 1 is Essential Maximum Entropy Based Tool for interpreting multiply-charged electrospray data).

### ACKNOWLEDGMENTS

This work was supported by the Foundation for Polish Science (FNP) programs co-financed by the European Union under the European Regional Development Fund [grant number POIR.04.04.00–00–3D8D/16–00], The National Centre for Research and

Development, and PrevEco supported by Norway grants in POLNOR2019 "Applied Research" program [grant number NOR/POLNOR/PrevEco/0021/2019].

## AUTHOR AFFILIATIONS

[1]International Institute of Molecular and Cell Biology in Warsaw, Warsaw, Poland
[2]Mossakowski Medical Research Institute Polish Academy of Sciences, Warsaw, Poland

## AUTHOR ORCIDs

Paweł Mitkowski ⓘ http://orcid.org/0000-0002-5422-5921
Elżbieta Jagielska ⓘ http://orcid.org/0000-0003-0925-8457
Izabela Sabała ⓘ http://orcid.org/0000-0002-5481-8671

## FUNDING

| Funder | Grant(s) | Author(s) |
|---|---|---|
| Fundacja na rzecz Nauki Polskiej (FNP) | POIR.04.04.00-00- 3D8D/16-00 | Paweł Mitkowski |
| | | Elżbieta Jagielska |
| | | Izabela Sabała |
| MNiSW \| Narodowe Centrum Badań i Rozwoju (NCBR) | NOR/POLNOR/PrevEco/ 0021/2019 | Paweł Mitkowski |
| | | Elżbieta Jagielska |
| | | Izabela Sabała |

## AUTHOR CONTRIBUTIONS

Paweł Mitkowski, Investigation, Methodology, Visualization, Writing – original draft | Elżbieta Jagielska, Funding acquisition, Project administration, Writing – original draft | Izabela Sabała, Conceptualization, Funding acquisition, Supervision, Writing – original draft, Writing – review and editing

## ADDITIONAL FILES

The following material is available online.

### Supplemental Material

**Supplemental material (Spectrum03546-23-S0001.docx).** Table S1; Fig. S1 to S3.

### Open Peer Review

**PEER REVIEW HISTORY (review-history.pdf).** An accounting of the reviewer comments and feedback.

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
