## [Reviewer comments · Microbiology Spectrum]

Microbiology Spectrum

Engineering of chimeric enzymes with expanded tolerance to ionic conditions

Paweł Mitkowski, Elżbieta Jagielska, and Izabela Sabala

Corresponding Author(s): Izabela Sabala, Mossakowski Medical Research Institute PAS

Review Timeline:

Submission Date:	October 2, 2023
Editorial Decision:	January 16, 2024
Revision Received:	February 28, 2024
Accepted:	March 26, 2024

Editor: Montarop Yamabhai

Reviewer(s): Disclosure of reviewer identity is with reference to reviewer comments included in decision letter(s). The following individuals involved in review of your submission have agreed to reveal their identity: Kiattawee Choowongkamon (Reviewer #2)

Transaction Report:

DOI: <https://doi.org/10.1128/spectrum.03546-23>

Re: Spectrum03546-23 (Engineering of chimeric enzymes with expanded tolerance to ionic conditions)

Dear Dr. Izabela Sabala:

Thank you for the privilege of reviewing your work. Below you will find my comments, instructions from the Spectrum editorial office, and the reviewer comments.

Please revise the manuscript as suggested by comments from the 3 reviewers, especially reviewer no.3, who has raised several valid concerns. Please submit point-by-point responses to all comments and provided the page and line numbers in the revised manuscript where you have addressed all these issues to help facilitate the reviewing process.

Revision Guidelines

Sincerely,
Montarop Yamabhai
Editor
Microbiology Spectrum

Reviewer #1 (Comments for the Author):

Work on grammatical errors and language construction and punctuation. For example, in your abstract, you wrote, "and as reported WHO is rising to dangerously high level".

The paper will benefit from language revision. Organisms mentioned should be italicised.

Reviewer #2 (Comments for the Author):

I would be nice if the author can show the 3D model of all 3 chimeric enzymes. Also discuss how 3 different SH3b binding domain can show different effect in term of structure and activity relationship.

Reviewer #3 (Comments for the Author):

1. Is there any good reasons that you do the experiments for number pf colonies killed and turbidity reductio assay together? If so, please explain clearly.
2. There is confusion in teminology such as low ionic condition. What do you mean by this? Is there difference between ionic strength and ionic condition? You had better specify the condition at each time.
3. Is there any reason you used 500 nM for enzyme activity assay and for heat stability assay, 1 μ M? Furthermore, there is no specification for the enzyme concentration in serum and milk and EDTA effect!!!
You had better explain the reason that you employed those concentration for the determination of enzyme activity.
4. There is absurd expression such as " To exclude the effect of ionic strength, all buffers had conductivity adjusted to" because you change the ionic strength with NaCl. Please find out more appropriate place to put this sentence in the manuscript and describe appropriately.

Dear Editor,

First of all, we would like to express our gratitude for the reviews that pointed out aspects missing in our manuscript. We truly believe that by following the suggestions of the reviewers, the manuscript has been improved. We hope that we addressed all the points raised by the reviewers and that the revised version of the manuscript will meet yours and the reviewers' expectations.

Below are detailed responses to the reviewers' comments:

Reviewer #1:

Work on grammatical errors and language construction and punctuation. For example, in your abstract, you wrote, "and as reported WHO is rising to dangerously high level". The paper will benefit from language revision. Organisms mentioned should be italicised.

Following the suggestion of the reviewer, the manuscript was improved by professional editing service. Improvements have been made in grammar, punctuation, and sentence structure throughout the article. The names of microorganisms are now in italics.

Reviewer #2:

I would be nice if the author can show the 3D model of all 3 chimeric enzymes. Also discuss how 3 different SH3b binding domain can show different effect in term of structure and activity relationship.

Three-dimensional models of our chimeras were prepared using the AlphaFold online tool and are presented in Figure S2 (Supplementary Materials). The crystallographic structure of lysostaphin, composed of the M23 catalytic domain and SH3b domain (PDB code 4LXC), was used as a template. After careful analysis of three SH3b domains' generated models aligned with the crystal structure, we believe that any relevant conclusion cannot be drawn. There is no apparent relationship between the SH3b binding domains predicted structures and proven different properties as a part of chimeric enzymes.

Reviewer #3:

1. Is there any good reasons that you do the experiments for number of colonies killed and turbidity reductio assay together? If so, please explain clearly.

We use two types of tests to investigate bacteriolytic activity of various enzymes: turbidity reduction assay (TRA) and bacterial cells' survival assay. Those tests show two distinct phenomena: TRA measures cell lysis upon treatment with the enzymes, observed as reduction of turbidity in OD600, whereas enumeration of bacterial cells is more precise and give direct answer about the efficiency of tested proteins in elimination of particular initial cells number. Results of both tests do not always correlate to each other. In some cases, we observe a significant drop in OD while the number of surviving cells remained very high, and opposite, while the OD was only slightly reduced, the number of cells dropped significantly. The discrepancy observed may be a consequence of limitations of both approaches, TRA illustrates cell lysis rate but only to certain level and does not show the number of cells being lysed. Colony counts help to prove that measured OD drop led to cell death. Additionally, survival cells number estimation also demonstrates antibacterial activity of studied enzymes, even if the toxicity is not a consequence of cell lysis, which in that case cannot be shown in TBA assay, we

can still measure its bactericidal effectiveness. We therefore use TRA as a routine fast and convenient test to indirectly measure the activity of peptidoglycan hydrolases, and to confirm its bactericidal effect, we perform killing assay. We have explained the choice of the assays in the manuscript to make it clear for the reader.

2. There is confusion in terminology such as low ionic condition. What do you mean by this? Is there difference between ionic strength and ionic condition? You had better specify the condition at each time.

We have standardized the vocabulary used throughout the entire manuscript, consistently employing the term "ionic strength".

3. Is there any reason you used 500 nM for enzyme activity assay and for heat stability assay, 1 μ M? Furthermore, there is no specification for the enzyme concentration in serum and milk and EDTA effect!!! You had better explain the reason that you employed those concentration for the determination of enzyme activity.

We used enzyme concentrations of 500 nM in all assays. This has been now corrected and clearly described in the Materials and Methods section. For each test, enzyme stocks were prepared at a concentration of 1 μ M and then mixed with the bacterial suspension in a 1:1 volume ratio, resulting in a final concentration of 500 nM. We have modified the manuscript to make it clear.

4. There is absurd expression such as " To exclude the effect of ionic strength, all buffers had conductivity adjusted to" because you change the ionic strength with NaCl. Please find out more appropriate place to put this sentence in the manuscript and describe appropriately.

In fact, this sentence was unfortunate. It has been corrected.

Re: Spectrum03546-23R1 (Engineering of chimeric enzymes with expanded tolerance to ionic conditions)

Dear Dr. Izabela Sabala:

The manuscript has been satisfactorily revised.

Your manuscript has been accepted, and I am forwarding it to the ASM production staff for publication. Your paper will first be checked to make sure all elements meet the technical requirements. ASM staff will contact you if anything needs to be revised before copyediting and production can begin. Otherwise, you will be notified when your proofs are ready to be viewed.

Sincerely,
Montarop Yamabhai
Editor
Microbiology Spectrum